

# The concentration of single-stranded DNA-binding proteins is a critical factor in recombinase polymerase amplification (RPA), as revealed by insights from an open-source system

Francisco Cordoba-Andrade[1], Antolin Peralta-Castro[1], Paola L. García-Medel[1], Eduardo Castro-Torres[1], Rogelio Gonzalez-Gonzalez[1], Atzimba Y. Castro-Lara[1], Josue D. Mora Garduño[1], Claudia D. Raygoza[1], Noe Baruch-Torres[1], Alejandro Peñafiel-Ayala[1], Corina Diaz-Quezada[1], Cesar S. Cardona-Felix[2], Fernando Guzman Chavez[3], Carlos H. Trasviña-Arenas[4], Rogerio R. Sotelo-Mundo[5], Beatriz Xoconostle-Cazares[6], Agustino Martínez-Antonio[7] and Luis Gabriel Brieba de Castro[1]

[1] Unidad de Genómica Avanzada, CINVESTAV-IPN, Irapuato, Gto, Mexico
[2] SECIHTI-Centro de Investigaciones Biológicas del Noroeste, La Paz, Baja California Sur, Mexico
[3] Facultad de Química, Departamento de Alimentos y Biotecnología, Universidad Nacional Autónoma de México UNAM, Mexico City, Mexico
[4] Cinvestav, Centro Envejecimiento, Mexico City, Mexico
[5] Laboratorio de Estructura Biomolecular, Centro de Investigacion en Alimentacion y Desarrollo, A. C, Hermosillo, Sonora, Mexico
[6] Departamento de Biotecnología y Bioingeniería, CINVESTAV-IPN, Mexico City, Mexico
[7] Unidad Irapuato, CINVESTAV-IPN, Irapuato, Guanajuato, Mexico

Corresponding author
Luis Gabriel Brieba de Castro,
luis.brieba@cinvestav.mx

## ABSTRACT

Recombinase polymerase amplification (RPA) facilitates rapid, exponential, isothermal nucleic acid amplification without the need for specialized equipment. Since its development in 2006, RPA has been widely applied to detect hundreds of RNA and DNA targets, spanning point-of-care diagnostics and agricultural uses. However, its reliance on pre-assembled commercial kits limits flexibility for customization. In this study, we introduce an open-source alternative to commercial RPA kits, utilizing purified, heterologously expressed proteins to circumvent the fixed molar ratios of proprietary systems. Our method incorporates enzymes from the bacteriophage T4 homologous recombination pathway—single-stranded binding protein (gp32), recombinase (UvsX), and mediator (UvsY)—along with Moloney murine leukemia virus (MMLV) reverse transcriptase with enhanced thermal stability, and Bst and Bsu DNA polymerases. We assessed the impact of buffer composition, reagent concentrations, and reaction temperature using synthetic SARS-CoV-2 genes. Notably, gp32 concentration and buffer composition emerged as critical factors in optimizing RPA performance. Using this tailored system, we demonstrated successful detection of the SARS-CoV-2 N gene on lateral flow devices (LFDs) with cDNA from eight clinical samples, achieving results consistent with RT-PCR. This open-source RPA platform provides an adaptable and cost-effective alternative for researchers, enabling
the exploration of diverse experimental conditions and offering a viable solution for those without access to commercial kits.

## INTRODUCTION

Recombinase polymerase amplification (RPA) is a nucleic acid amplification method widely applied in point-of-care diagnostics for the detection of pathogens in humans, animals, and plants (*El-Tholoth et al., 2019*; *Hu et al., 2019*; *Piepenburg et al., 2006*; *Ranjan et al., 2020*; *Zhao et al., 2018*). RPA shares conceptual similarities with homologous recombination processes observed in bacteriophages, bacteria, and eukaryotes (*Daley et al., 2014*; *Lee, Gao & Yang, 2015*; *Piepenburg et al., 2006*). During RPA, an excess of single-stranded DNA binding protein, such as bacteriophage T4 gp32 or bacterial single-stranded binding (SSB), stabilizes a long oligonucleotide homologous to the target region. A mediator protein, (*i.e.,* bacteriophage T4 UvsX), then facilitates the displacement of the SSB protein from the oligonucleotide and promotes the loading of a cognate recombinase (*Piepenburg et al., 2006*). Importantly, UvsY is necessary for destabilizing gp32 filaments by inducing DNA bending and promoting the exchange of gp32 for UvsX recombinase molecules, thereby enhancing the formation and stabilization of UvsX presynaptic filaments. UvsY assumes a heptameric open barrel conformation that encircles ssDNA and forms a complex with gp32-ssDNA in a 1:1 stoichiometry. This interaction not only disrupts gp32 binding to ssDNA but also facilitates the replacement of gp32 with UvsX, thereby amplifying UvsX activity in homologous recombination and DNA replication. The proposed model posits that UvsY induces a helical conformation in ssDNA, which inhibits gp32 binding and triggers the assembly of the ssDNA-UvsX filament, ultimately ensuring the efficient progression of recombination and replication processes (*Bleuit et al., 2004*; *Bleuit et al., 2001*; *Gajewski et al., 2016*; *Henry & Henrikus, 2021*). Homologous recombination initiates with the recombinase (*i.e.,* bacteriophage T4 UvsX or bacterial RecA) binding to a synthetic single-stranded oligonucleotide, forming a recombinase-ssDNA complex. This complex searches for homologous regions within double-stranded DNA and initiates strand invasion at homologous sites. Once the recombinase dissociates from the complex, a strand-displacing DNA polymerase, such as *Bacillus subtilis* DNA polymerase I, binds to the 3′-end of the primer and extends the DNA chain in the presence of dNTPs (*Kiefer et al., 1998*). This amplification cycle repeats exponentially, selectively amplifying a DNA sequence when two oligonucleotides are oriented in opposite directions on the double helix. Furthermore, RPA can be combined with reverse transcription (RT-RPA) to enable the detection of single- and double-stranded RNA viruses.

The amplified DNA product of the RPA reaction is typically detected using a lateral flow immunoassay device (LFD). In this combined LFD-RPA assay, visualization is enabled by a probe containing an abasic site, a 3′ end terminator, and FAM/FITC/digoxigenin-labeled

5′ termini. Upon hybridization of the probe to the amplified nucleic acid, the substrate is cleaved at the abasic site by the APE1 enzyme, generating a new 3′ end for subsequent detection (*El-Tholoth et al., 2019*). Most RPA-based diagnostic assays utilize enzymes derived from bacteriophage T4, provided in pre-assembled RPA kits (*Burkhardt et al., 2019*; *Hu et al., 2019*; *Lobato & O'Sullivan, 2018*). Recent studies have made progress in using and optimizing individually expressed bacteriophage T4 proteins for RPA-based diagnostics (*Juma et al., 2021*; *Kojima et al., 2021*). However, these studies still rely on commercially available enzymes for nucleic acid amplification and energy regeneration. Unlike other amplification techniques such as PCR, LAMP, or multiple displacement amplification (MDA), where open-source enzymatic components are widely available, RPA lacks a comprehensive open-source system (*Cerda et al., 2024*; *Spits et al., 2006*). In this study, we present a protocol for purifying all the essential components for RPA, including homologous recombination proteins, energy regeneration systems, DNA and RNA amplification enzymes, and a lateral flow immunoassay for detection. Our findings indicate that the ratio of the T4 homologous recombination components and the use of Bsu DNA polymerase are critical for successful RPA.

## MATERIALS & METHODS

### Synthetic gene assembly of bacteriophage T4 homologous recombination components and strand-displacement DNA polymerases

The DNA sequences encoding UvsX, UvsY, and gp32 were synthetically synthesized and optimized for bacterial expression (Twist Bioscience). These genes were subcloned into pCRI1b vectors using Nde I and BamH I restriction sites (*Goulas et al., 2014*) and into modified pET19b vector, where the thrombin cleavage site was replaced with a PreScission protease (pps) site. The sequences of these optimized genes are provided in the Supplementary Materials, and details of the cloning enzymes used are outlined in Table S1. The gene segment encoding the large fragment of *Bst* DNA polymerase was codon-optimized (Twist Bioscience) and subcloned into a pCOLD I vector (Takara), while the *Bsu* DNA polymerase gene was obtained from Addgene (Plasmid # 163911) courtesy of Professor Paul Freemont (*Patchsung et al., 2023*). The heat-resistant reverse transcriptase variant MM4 from Moloney murine leukemia virus was kindly provided by Professor Kiyoshi Yasukawa (*Okano et al., 2017*; *Yasukawa et al., 2010*). The plasmids encoding the protein products used in this study have been deposited in Addgene as follows: pET19b-pss-gp32 (Plasmid #236044), pET19b-pps-UvsY (Plasmid #236045), pET19b-pps-UvsX (Plasmid #236046), pCOLDI-KF-BstDNApolI (Plasmid #236047), and pET19b-pps-EndoIV from *Thermus thermophilus* (Plasmid #236048).

### Purification of RPA components

Recombinant proteins were expressed in *Escherichia coli* BL21 (*DE3*) co-transformed with the pKJ-E7 chaperone plasmid (Takara) and purified according to standard protocols for DNA-binding proteins, as detailed in Supplementary Materials (*Baruch-Torres & Brieba, 2017*; *Meneses et al., 2010*; *Peralta-Castro, Baruch-Torres & Brieba, 2017*). The purified

enzymes include UvsX, UvsY, gp32, *Bst* DNA polymerase, creatine kinase, thermostable MMLV reverse transcriptase (M4-MMLV-RT), and thermoresistant bacterial abasic site endonuclease (*nfo*). Importantly, proteins were stored in a buffer contained only 10% of glycerol and were subject to snap freezing in liquid nitrogen. Before conducting any biochemical assays, the enzymes were diluted in reaction buffer without glycerol. We calculated that the final concentration of glycerol during the RPA reaction is less than 1%. As previously reported by Kojima and Morina, glycerol has an inhibitory effect on RPA (*Kojima et al., 2021*; *Morimoto et al., 2024*). All RPA-associated enzymes were subject to an *in vitro* assay using fluorescent labeled 3′ and 5′ ssDNA oligonucleotide to assure that are free of contaminating nucleases, following reported procedures (*Penafiel-Ayala et al., 2024*). Protein concentration was measured using a Bio-Rad protein assay curve with BSA as standard on a 96-well plate (*Bradford, 1976*).

## Cloning of test DNA and RNA

To evaluate the T4 homologous recombination (HR) system for RPA, we utilized a synthetic DNA construct encoding segments of the SARS-CoV-2 RNA polymerase (RdRp), envelope (E), and nucleocapsid (N) genes. We synthesized this DNA segment aimed to harbor the DNA segment that is amplified in the "Berlin protocol", developed by the Institute of Virology at Charité Hospital in Berlin, as this protocol targets the RdRp, E, and N genes of SARS-CoV-2, utilizing the reference sequence NC045512-2 (Wuhan-185 Hu-1) (*Corman et al., 2020*). The human RNAPase gene served as a positive control. We synthesized regions of these genes containing the relevant sequences for SARS-CoV-2 in accordance with the Berlin Protocol. This DNA segment was then cloned between the Nco I and BamH I restriction sites in the pET28b vector, which confers kanamycin resistance. Two T7 RNA polymerase promoter sequences were incorporated upstream of the E and RNAPase genes within the synthetic construct. A detailed description of the cloned SARS-CoV-2 genes can be found in the Supplementary Materials.

## T4 HR assays

For D-loop assembly utilizing T4 homologous recombination proteins, we employed a 90 nt oligonucleotide (5′Cy5-TTT TGC AAA AGA AGT TTT GCC AGA GGG GGT AAT AGT AAA ATG TTT AGA CTG GAT AGC GTC CAA TAC TGC GGA ATC GTC ATA AAT ATT CAT-3′) that exhibits perfect homology to the supercoiled M13mp18 RFI plasmid/bacteriophage (7,249 bp). D-loop assembly reactions (10 µL) were conducted at 37 °C in a buffer containing 50 mM Tris-HCl (pH 7.5), three mM DTT, 1 mM $MgCl_2$, two mM ATP, 0.1 mg/mL BSA, 2% glycerol, 12 mM phosphocreatine, and one µM creatine kinase. The 90-mer Cy5-oligo was incubated with 300 nM EcRecA or 600 nM UvsX for 10 min to facilitate filament formation. In reactions involving only gp32 or UvsY, the concentrations used were 600 nM and six µM, respectively. To initiate D-loop formation, 1 nM of supercoiled M13mp18 RFI plasmid was added, and the $MgCl_2$ concentration was increased to 10 mM. The reactions were incubated for 8 min, followed by the addition of 3% SDS to halt the reaction. The reaction products were then mixed with 2X loading buffer (25 mM Tris-HCl, pH 7.5, 8% glycerol) and analyzed *via* electrophoresis in a 0.8%

agarose gel using 1X TAE buffer at 75 V for 90 min, followed by staining with ethidium bromide. The gel was visualized using the Amersham Typhoon scanner.

For the D-loop extension assay, 10 µL reactions were conducted using 600 nM of UvsX, gp32, and UvsY to assemble the D-loop as previously described. The reactions included 80 nM each of *Bsu* and *Bst* polymerases and 100 µM of dNTPs. Incubation times of 3, 10, and 30 min were employed, and products were analyzed by electrophoresis and visualized as previously detailed.

## Homemade RPA system or open RPA system

To set up the RPA systems we used a set of previously designed oligonucleotides based on the Berlin protocol (https://www.ncbi.nlm.nih.gov/nuccore/NC_045512) as follows:

| | |
|---|---|
| N FwPrimer | TTTGGTGGACCCTCAGATTCAACTGGCAGTAAC |
| N Rv Primer | /5Biosg/GAATTTAAGGTCTTCCTTGCCATGTTGAGTGAG |
| Rv Primer-EneN | GAATTTAAGGTCTTCCTTGCCATGTTGAGTGAG |
| | |
| Polymerase_FwPrimer | AAGTATTGAGTGAAATGGTCATGTGTGGCGGTTC |
| Polymerase_Rv Primer | /5Biosg/GACATACTTATCGGCAATTTTGTTACCATCAGT |
| Polymerase_Rv Primer | GACATACTTATCGGCAATTTTGTTACCATCAGT |
| | |
| GEN_Epsilon FwPrimer | TCGGAAGAGACAGGTACGTTAATAGTTAATAGCGT |
| GEN_Epsilon LwPrimer | /5Biosg/TTTACAAGACTCACGTTAACAATATTGCAGCAG |
| GEN_Epsilon LwPrimer | TTTACAAGACTCACGTTAACAATATTGCAGCAG |

The oligonucleotides used for amplifying the N, RdRp, and E genes corresponded to regions of 150, 179, and 143 nucleotides, respectively. RPA reactions were carried out in a buffer consisting of 50 mM Tris-HCl (pH 8.0), two mM DTT, 40 mM phosphocreatine, 240 µM dNTPs, 5.5% PEG 35,000, and one µM of each oligonucleotide. Initially, the concentrations of UvsX, UvsY, and gp32 were 5.9, 0.18, and 7.8 µM, respectively, while *Bsu* and *Bst* DNA polymerases were included at 0.62 µM, and creatine kinase at 1.2 µM. Plasmid DNA was added at a concentration of 0.125 ng in a total reaction volume of 10 µL. Reactions were initiated by the addition of a magnesium acetate solution to a final concentration of 14 mM, and incubated at various temperatures ranging from 37 to 57 °C. After 1 h of incubation, the reactions were analyzed using 1.5% agarose gel electrophoresis.

## Lateral flow immunoassays for evaluation of the RPA reaction using SARS-CoV-2 clinical samples

The study was conducted according to the guidelines of the Declaration of Helsinki and approved by the ETHICS COMITE FOR RESEARCH IN HUMAN BEINGS (COBISH-CINVESTAV) (Protocol number: 062/2020). Written, informed consent was signed by each participant and this study excluded the participation of minors. cDNA samples of healthy and infected individuals with SARS-CoV-2 were generously provided from a previous study (*Tapia-Sidas et al., 2023*). Saliva samples were collected from volunteers seeking SARS-CoV-2 testing in México City during September 2021. RNA was extracted from these samples using the Biopure Self-Sampling Kit (Biopure, CDMX, México)

according to the manufacturer's instructions and quantified with a NanoDrop One (Thermo Scientific, Waltham, MA, USA) as previously described (*Tapia-Sidas et al., 2023*). The purified RNA was then used for cDNA synthesis with the SuperScript III™ First-Strand Synthesis System (Invitrogen, Waltham, MA, USA), following the manufacturer's protocol. Lateral flow immunoassays were conducted by amplifying the SARS-CoV-2 N gene. This amplification used a set of three oligonucleotides, including 0.5 μM of the Nfo probe (5′-GCGATCAAAACAACGTCGGCCCCAAGGTTTACC/IdSp/AATAATACTGCGTCT-3′), along with one μM of the NFw and NRw oligonucleotides RPA reactions were executed in 50 mM Tris-HCl (pH 8.0), two mM DTT, 40 mM phosphocreatine, 240 μM dNTPs, 5.5% PEG 35000. UvsX, UvsY, gp32, APE1 endonuclease, *Bsu* DNA polymerase, and creatine kinase were 5.9, 0.18, 7.8, 0.5 μM. at 0.62 and 1.2 μM, respectively. Reactions were incubated at 42 °C for 10 min, and RPA products were run on a 2% agarose gel. A sample was diluted 1:50 in running buffer (Milenia Biotec, Gießen, Germany) and 10 μL of this dilution was loaded onto a lateral flow test strip (Milenia Biotec). The strip was placed in a crystallography plate with running buffer (Milenia Biotec) and the strip was visualized after 5 min.

## RESULTS

### Recombinantly expressed T4 Homologous Recombination proteins assemble a D-loop on plasmid DNA

In RPA, the T4 single-stranded binding protein gp32 coats single-stranded DNA that acts as a primer, while the mediator protein UvsY displaces gp32 and facilitates the binding of the UvsX recombinase to the single-stranded DNA (Fig. 1A). This base-pairing process leads to the formation of a displacement loop, or "D-loop", which is a heteroduplex product where a single-stranded DNA with a free 3′-OH hybridizes to a homologous region of double-stranded DNA (dsDNA). RPA employs a DNA polymerase with strand-displacement capabilities to extend the presynaptic filament acting as a primer (*Alberts et al., 1975*; *Bleuit et al., 2001*; *Gajewski et al., 2016*; *Liu & Morrical, 2010*; *Piepenburg et al., 2006*) (Fig. 1A).

To create an open-source RPA enzyme system, we constructed a minimal RPA system utilizing codon-optimized genes for the bacterial expression of the homologous recombination (HR) apparatus from bacteriophage T4. These proteins can be purified to homogeneity as histidine-tagged proteins (Table S1), following established protocols (*Formosa & Alberts, 1986*; *Gajewski et al., 2016*; *Hosoda & Moise, 1978*; *Juma et al., 2021*; *Juma et al., 2022*; *Kojima et al., 2021*). All components of the T4 HR system were purified with yields exceeding one mg per liter of bacterial culture and demonstrated no nuclease activity after three chromatographic purification steps (using IMAC, heparin, and ionic exchange columns) (Tables S1 and S2, Fig. 1B). The UvsY and gp32 proteins were assessed as His-tagged proteins. The N-terminal tag does not affect the function of UvsY, and the protein-protein interactions of gp32 are localized to its C-terminal tail (*Juma et al., 2022*; *Perumal, Nelson & Benkovic, 2013*). While *Piepenburg et al. (2006)* solved the issue of placing an N-terminal histidine tag on UvsX, that appears to negatively affect its enzymatic, by using a C-terminal histidine tag, we solved this problem by removing the histidine tag by the action of the enzyme PreScission Protease.

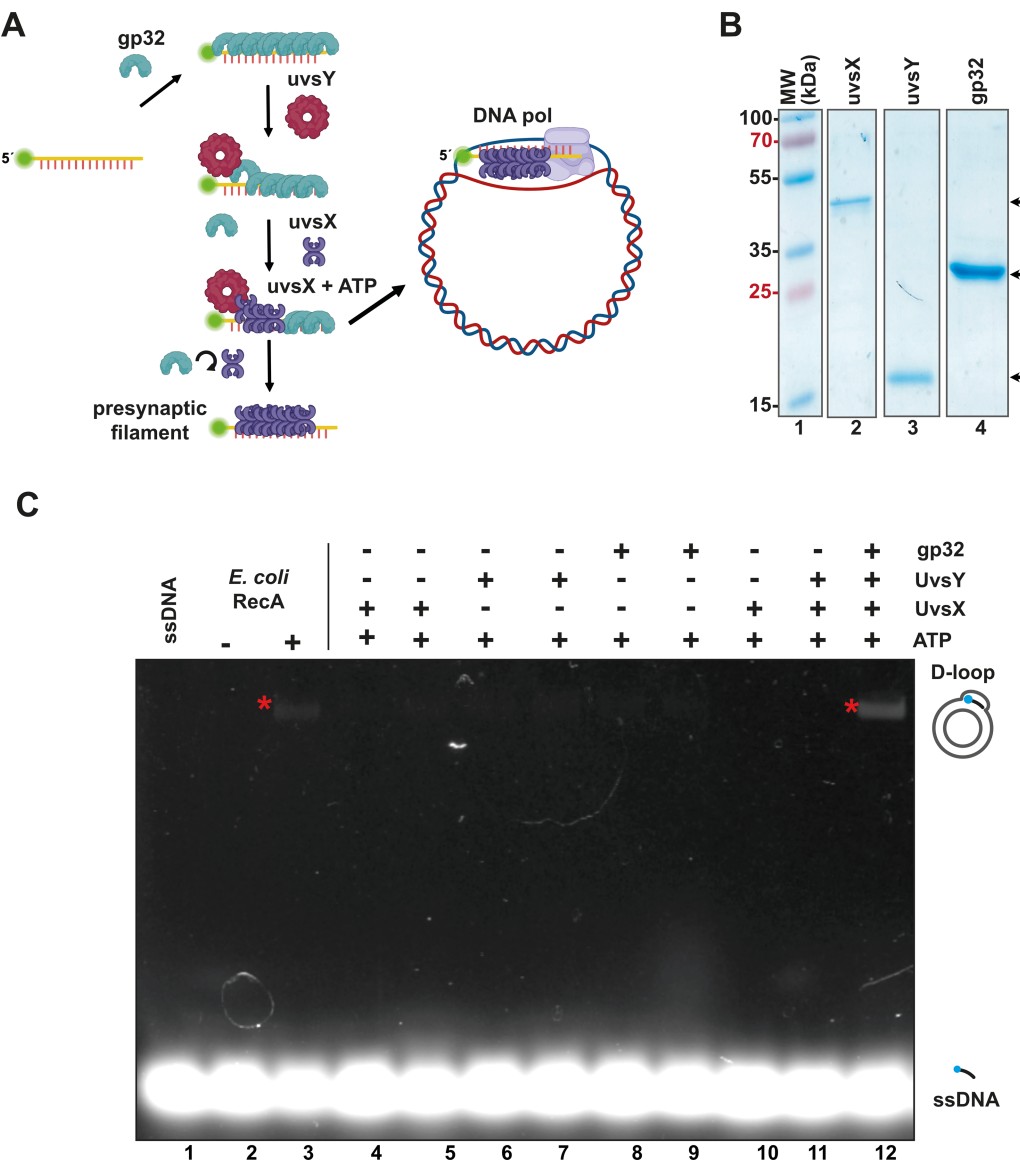

**Figure 1** **The recombinantly expressed T4 homologous recombination (HR) system successfully assembled a D-loop.** (A) Schematic representation of T4 HR D-loop. During this process, single-stranded DNA (ssDNA) is coated with gp32, while the mediator protein UvsY displaces gp32 from the ssDNA and facilitates the loading of the recombinase UvsX. The loading of UvsX promotes the formation of presynaptic filaments that search for homologous regions within large double-stranded DNA (dsDNA) and execute base pairing with the ssDNA. The 3′-OH of the ssDNA oligonucleotide serves as a primer for DNA polymerases. (Panel created with BioRender.com). (B) A total of 12.5% SDS-PAGE of purified T4 HR proteins is presented. The gel demonstrates the purity and molecular weight of gp32, UvsX, and UvsY, with their relative migration indicated by arrows. (C) D-loop assembly mediated by T4 HR components was compared to that of bacterial RecA. Bacterial RecA, at a concentration of 300 nM, annealed a Cy5-labeled DNA substrate at two nM to a complementary region of a supercoiled pGEM plasmid in the presence of 5 mM ATP (lanes 2 and 3). The Cy5-labeled DNA substrate was unable to self-anneal to the supercoiled template (lanes 4 and 5). D-loop formation was assessed using two concentrations of the individual components of the T4 HR system in the presence of ATP (lanes 4 to 11). Reactions containing gp32, UvsX, and UvsY were performed at concentrations of 600 nM (lanes 4, 6, 8) or six μM (lanes 5, 7, 9, 10, 11, 12).

To evaluate the enzymatic activity of our purified T4 recombination system, we examined the formation of a D-loop on supercoiled plasmid DNA (Fig. 1C). D-loop formation was assessed using a 5′-labeled Cy-5 90-mer oligonucleotide homologous to the M13mp18 plasmid. The hybridization of the labeled oligonucleotide was tested alongside two concentrations (with a ten-fold difference between them) of the individual proteins gp32, UvsX, and UvsY in the presence of ATP (Fig. 1C, lanes 4 to 11). Notably, D-loop assembly did not occur when only one of the individual T4 HR components was present (Fig. 1C, lanes 4 to 11). This contrasts with bacterial RecA, which can assemble D-loops without accessory proteins (*McIlwraith et al., 2001*) (Fig. 1C, lane 3). D-loop assembly was successfully achieved only when gp32, UvsX, and UvsY were all present in the reaction along with ATP (Fig. 1C, lane 12).

### *Bsu* and *Bst* DNA polymerases execute primer extension coupled with strand-displacement on UvsX-primer filaments

In addition to the enzymes involved in T4 homologous recombination, RPA necessitates a strand-displacement DNA polymerase, creatine kinase for ATP regeneration, a thermostable reverse transcriptase, and, in specialized applications, a DNA glycosylase with lyase activity to cleave abasic sites or uracil-containing oligonucleotides. In this study, we obtained or subcloned these enzymes (Fig. 2A). Several DNA polymerases can perform strand displacement, including those from bacteriophage phi29 and from *Staphylococcus aureus* (*Sau*), *Bacillus subtilis* (*Bsu*), and *Bacillus stearothermophilus* (*Bst*). The DNA polymerases utilized in this research are modified versions of full-length *Bsu* and *Bst* DNA polymerases. These full-length enzymes contain three domains: a 5′-3′ exonuclease domain, an inactive 3′-5′ exonuclease domain, and a 5′-3′ polymerization domain. However, the version of these DNA polymerases most commonly used in research and biotechnological applications corresponds to the Klenow fragment, which retains the 5′-3′ polymerization domain but lacks the 3′-5′ exonuclease activity. In this work, we refer to the Klenow fragments of the full-length enzymes as *Bst* DNApolI and *Bsu* DNApolI. Notably, *Bsu* DNApolI and *Bst* DNApolI exhibit 67% amino acid identity in their Klenow fragment domains. During our research, the initial construct expressing *Bst* DNApolI was largely insoluble (*Chim et al., 2018*). To address this issue, we codon-optimized and subcloned the gene into a pCOLD I vector (Takara) aiming to enhance its solubility using a low induction temperature. Both DNA polymerases were purified through three chromatographic steps and concentrated to five mg/mL, maintaining activity for at least two years at −20 °C (Fig. 2A and Supplementary Data). To evaluate D-loop extension by these DNA polymerases, we utilized D-loops assembled with gp32, UvsY, and UvsX (Fig. 2B, lane 2). Both *Bsu* DNAP and *Bst* DNAP were able to extend these primers within the first 3 min of incubation, successfully synthesizing one complete turn of the M13mp18 circular plasmid (Fig. 2B, lanes 3 to 10). However, both *Bsu* DNAP and *Bst* DNAP encountered challenges in further amplifying the M13mp18 plasmid after one complete round of amplification. Notably, extension reactions with *Bsu* DNAP yielded amplification products that exceeded one complete round of amplification at 10 and 30 min of incubation (Fig. 2B, lanes 5 and 6).

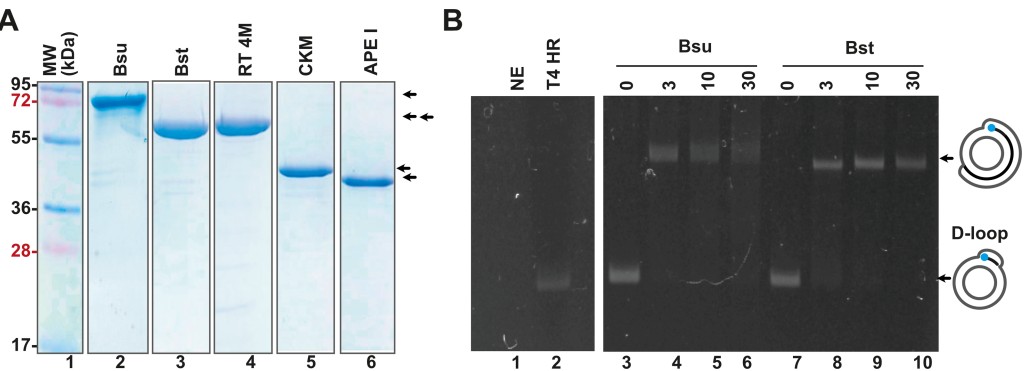

**Figure 2** **Open-source RPA system.** (A) SDS-gel showing purified *Bsu* DNA pol, *Bst* DNA pol, RT 4M, Chicken creatine kinase, and human APE1 An arrow indicates the identity of each protein. (B) D-loop extension using a supercoiled M13mp18 plasmid at a concentration of 1 nM. Agarose gel showing the use of a D-loop assembled with T4 HR proteins (lane 2) as a primer for *Bsu* and *Bst* DNA polymerases (80 nM) (lanes 3 to 6 and 7 to 10, respectively) during a timer course from 0 to 30 min of incubation. The full-length extension product of the M13mp18 plasmid is indicated by an arrow.

## RPA isothermal amplification of SARS-CoV2 genes using gp32, UvsY, and UvsX

To validate the RPA reaction, we constructed synthetic DNA sequences encoding segments of the SARS-CoV-2 RNA polymerase (RdRp), envelope (E), and nucleocapsid (N) genes, as well as the human RNAPase gene. Using specific RPA oligonucleotides, we aimed to amplify regions measuring 179 bp, 143 bp, and 150 bp for the RdRp, E, and N genes, respectively (Fig. 3A). A control experiment employing conventional PCR with both 5′ biotinylated and unmodified primers yielded amplification bands corresponding to the expected molecular weights (Fig. 3B). For the RPA experiment, we utilized DNA and enzyme concentrations consistent with those reported by *Piepenburg et al. (2006)*, and we compared the efficiency of these reactions to conventional PCR amplification (Fig. 3C). In these studies, we used all proteins, with the exception of UvsX as N-terminal histidine-tagged proteins. The histidine tag of UvsX was proteolytically removed and only harbors a Gly, a Pro and a His before the N-terminal Met. The RPA reactions were incubated in the presence of ATP and creatine kinase as an ATP-regeneration system, initiating the reaction with the addition of metal ions. Chicken creatine kinase was sourced from Addgene, with generous support from Professor Ueda (Data S1) (*Shimizu et al., 2001*). We tested our RPA reactions in the presence and absence of 80 mM of potassium acetate (CH3COOK) as the presence of this salt stimulates RPA activity mediated by T4 HR enzymes at 37 °C (*Juma et al., 2021*; *Kojima et al., 2021*) and this salt at 100 mM is a component of the reaction buffer originally proposed by *Piepenburg et al. (2006)*.

We also assayed the effect of 5′ biotynated and unmodified primers on the reaction.

In our RPA reactions utilizing primers specific to the RdRp, E, and N genes, we observed the presence of RPA-generated DNA bands corresponding to the 179 bp and 150 bp products for the RdRp and N genes, respectively, amplifying alongside conventional PCR products (Fig. 3C, lanes 4, 6, and 16). The intensity of the RPA products was notably

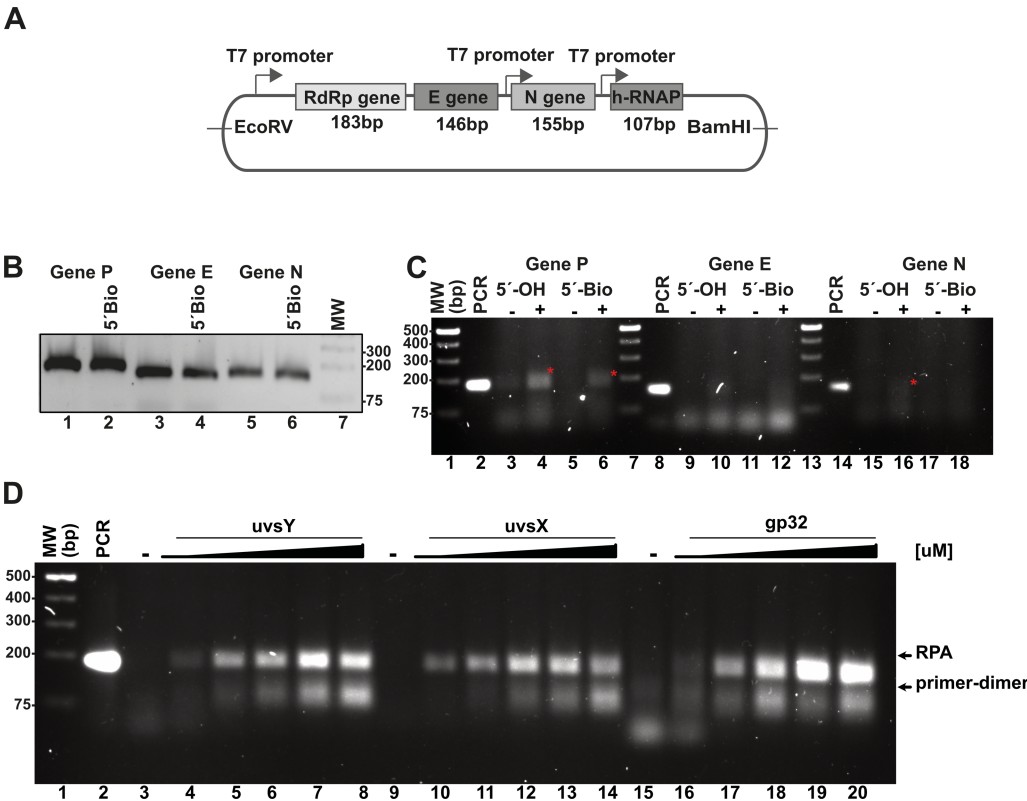

**Figure 3** *In vitro* **system used to evaluate RPA based on the isothermal amplification of SARS-COV2 genes.** (A) Diagram indicating the synthetic SARS-CoV-2 DNA cloned in a pET28b plasmid and the number of amplified base pairs at a specific pair of RPA oligonucleotide would amplify for each gene. The diagram also depicts the RNA promoter sequences that facilitate the generation of SARS-CoV-2-like transcripts. (B) PCR amplification results for the P, E, and N genes utilizing both free oligonucleotides and 5′-biotinylated oligonucleotides. (C) RPA reactions targeting the P, E, and N genes were conducted both in the absence and presence of 40 mM potassium acetate, using 5′-OH and 5′-biotinylated oligonucleotides. The amplified RPA products are marked with a red asterisk. (D) Assessment of the impact of varying concentrations of T4-HR proteins on RPA efficiency. RPA reactions were performed with increasing concentrations of UvsY (0, 0.47, 0.94, 1.88, 2.82, 3.76 μM), UvsX (0, 0.6, 1.35, 2.7, 4, 5.4 μM), and gp32 proteins (0, 3.33, 6.6, 13.33, 20, 26.6 μM). The relative migration of the RPA products and primer-dimers is indicated by arrows. In each standard RPA reaction, the concentration of one component of the T4-HR system was increased while maintaining the other two proteins at constant levels.

enhanced in the presence of potassium acetate (Fig. 3C, lanes 4 and 6; lanes 16 and 18). Conversely, for reactions employing primers specific to the E gene, we detected faint amplification products with lower molecular masses than the expected PCR product. These findings suggest a differential efficiency of primers in RPA reactions, indicating that those designed for RPA amplification are more readily utilized (*Higgins et al., 2019*). Moreover, our data reveal a propensity for the RdRp and E gene primers to self-anneal, leading to the formation of low molecular weight amplification products or primer-dimers (Fig. 3C, lanes 13 and 14). The use of software for designing RPA primers and evaluating a variety

of primer options is critical for successful RPA, particularly to minimize the formation of primer-dimer products.

## The stoichiometry between the T4-HR components is critical for RPA isothermal amplification

RPA based methods employ extremely different concentrations of T4-HR proteins (*Juma et al., 2022*; *Kojima et al., 2021*; *Piepenburg et al., 2006*). For example, Kojima and coworkers use T4-HR concentration of 12.7 and 4.6 µM of UvsX and gp32, respectively, whereas Piepenburg and coworker use 2.7 and 26.8 mM concentrations for both proteins (*Juma et al., 2022*; *Kojima et al., 2021*; *Piepenburg et al., 2006*). Comparing the molar differences between gp32, UvsX, and UvsY, the variations can be more than 20-fold among the different protocols (Table S3). To evaluate the impact of varying T4-HR protein concentrations, we systematically increased the concentration of one T4-HR protein while maintaining the concentrations of the other two constant (Fig. 3D). Our observations indicate that RPA reactions exhibit sensitivity to the relative concentrations of UvsY and, particularly, gp32 (Fig. 3D, lanes 4 to 8 and 16 to 20). Conversely, the yield of RPA products remained consistent with increasing concentrations of UvsX. Based on these findings, we chose to conduct reactions with the following T4-HR protein concentrations: 27 µM for gp32, 1.8 µM for UvsY, and 2.7 µM for UvsX. Notably, reactions employing a four-fold increase in gp32 concentration resulted in at least a two-fold enhancement in RPA product yield (Fig. 3D).

## Effect on the temperature and pH on RPA efficiency

Although we were able to generate a robust RPA product using our T4-HR systems, we also observed the presence of an amplification band of lower molecular mass, likely indicative of primer-dimer formation due to the primers' propensity to self-anneal (Fig. 3D). To address this issue, we aimed to investigate the influence of temperature, pH, and additives, as well as their optimal application for *Bsu* or *Bst* DNA polymerases. Specifically, we examined whether increasing the reaction temperature would mitigate or reduce the occurrence of non-specific amplification products. Literature on RPA amplification suggests that effective reactions are typically conducted at temperatures ranging from 25 °C to 42 °C (*Davi et al., 2019*; *Lillis et al., 2014*; *Oscorbin & Filipenko, 2023*). For RPA to be effective, the reaction temperature must remain below the denaturation temperature of the RPA enzymatic components and the melting point of the specific oligonucleotides. The original RPA protocol proposed by Piepenburg and coworkers was conducted at 37 °C a temperature in which T4 HR maybe optimal (*Piepenburg et al., 2006*). However, at this specific temperature our RPA oligonucleotides most probably self-anneal generating primer-dimers and other nonspecific amplification products. To circumvent this problem, we evaluated RPA reactions at three different pHs (7.5, 8.0, and 8.5) and temperatures (42 °C, 47 °C, and 52 °C). A 5 °C increase from 37 °C to 42 °C successfully eliminated the formation of primer-dimer products, regardless of whether *Bsu* or *Bst* DNA polymerases were employed (Fig. 4A, lanes 3, 6, 9, 12, 15, 18). While RPA was feasible at 47 °C, we observed a smear accompanying the RPA amplification band, which was absent at 42 °C

(Fig. 4A, lanes 4, 7, 10). Our findings suggest that T4 HR-mediated RPA can be effectively performed at temperatures ranging from 42 °C to 47 °C. Based on these results, we concluded that 42 °C is the optimal temperature for the RPA process using our specific set of primers. The substantial decrease in activity at temperatures between 47 °C and 52 °C for reactions harboring *Bsu* or *Bs*t DNA polymerases indicates potential inactivation of the enzymes involved in T4 HR. Notably, the melting temperatures for Bsu and *Bst* DNA polymerases are 54 °C and 72 °C, respectively (Table S4), while the optimal activity range for *Bst* DNA polymerase is between 45 °C and 65 °C (https://www.neb.com/) (*Kiefer et al., 1998*). The observed lack of activity at 52 °C is likely attributed to the thermal inactivation of UvsX, gp32, or UvsY, as mesophilic bacterial RecA proteins maintain stability up to 42 °C. To support this hypothesis, we assessed the melting temperatures of UvsX, UvsY, and gp32, revealing that UvsX and gp32 have melting temperatures of 49 °C. This finding aligns with the observed absence of isothermal amplification at 52 °C (Fig. 4A) and corroborates previously reported thermal stability measurements for gp32 (*Pant et al., 2004*). Collectively, our analysis is consistent with findings from commercial RPA systems, which operate effectively at temperatures up to 42 °C, and can extend to 45 °C, but not 50 °C (*Yang et al., 2020*). Furthermore, our results underscore the critical importance of employing effective primer-design strategies and the evaluation of multiple primers and temperatures to mitigate amplification artifacts (*Boyle et al., 2014*; *Higgins et al., 2019*; *Koressaar et al., 2018*; *Ma et al., 2017*; *Yang et al., 2021*).

## Effect of additives and pH on RPA reactions catalyzed by *Bsu* and *Bst* DNApol

Enzymatic reactions can be enhanced by various factors, including salt, detergents, crowding agents, and osmolytes. In this study, we evaluated the specific effects of DMSO, betaine, trehalose, Triton-X, and sorbitol on RPA efficiency at three different pH levels (7.5, 8.0, and 8.5), utilizing concentrations of additives previously shown to promote RPA or PCR DNA synthesis. Our results indicate that the addition of DMSO, betaine, and Triton-X significantly increased RPA efficiency at pH 8.5. Notably, for the N gene, the incorporation of 4% DMSO enhanced reaction efficiency, which contrasts with its previously reported inhibitory effects (*Kojima et al., 2021*). DMSO is known to improve PCR and LAMP reactions, likely by inhibiting the formation of secondary structures and reducing non-specific annealing, thereby facilitating the synthesis of longer products (*Kang, Lee & Gorenstein, 2005*; *Kim et al., 2023*).

## Detection limits of the T4 RPA assay

Reported RPA detection limits using commercial methods coupled to lateral flow or CRISPR can be as low 10 molecules per reaction (*Tan et al., 2022*; *Wang et al., 2018*). T4 HR proteins without a coupled method can detect up to 60 molecules of DNA (*Juma et al., 2022*). To determine the detection limits of our open-source RPA system, we use our SARS-CoV2 DNA segment of two kb cloned into a pET28b plasmid and the optimized reactions, including the addition of DMSO, potassium acetate, an optimal temperature of 42 °C and elevated molar concentration of gp32 in the reaction mixture. We started the

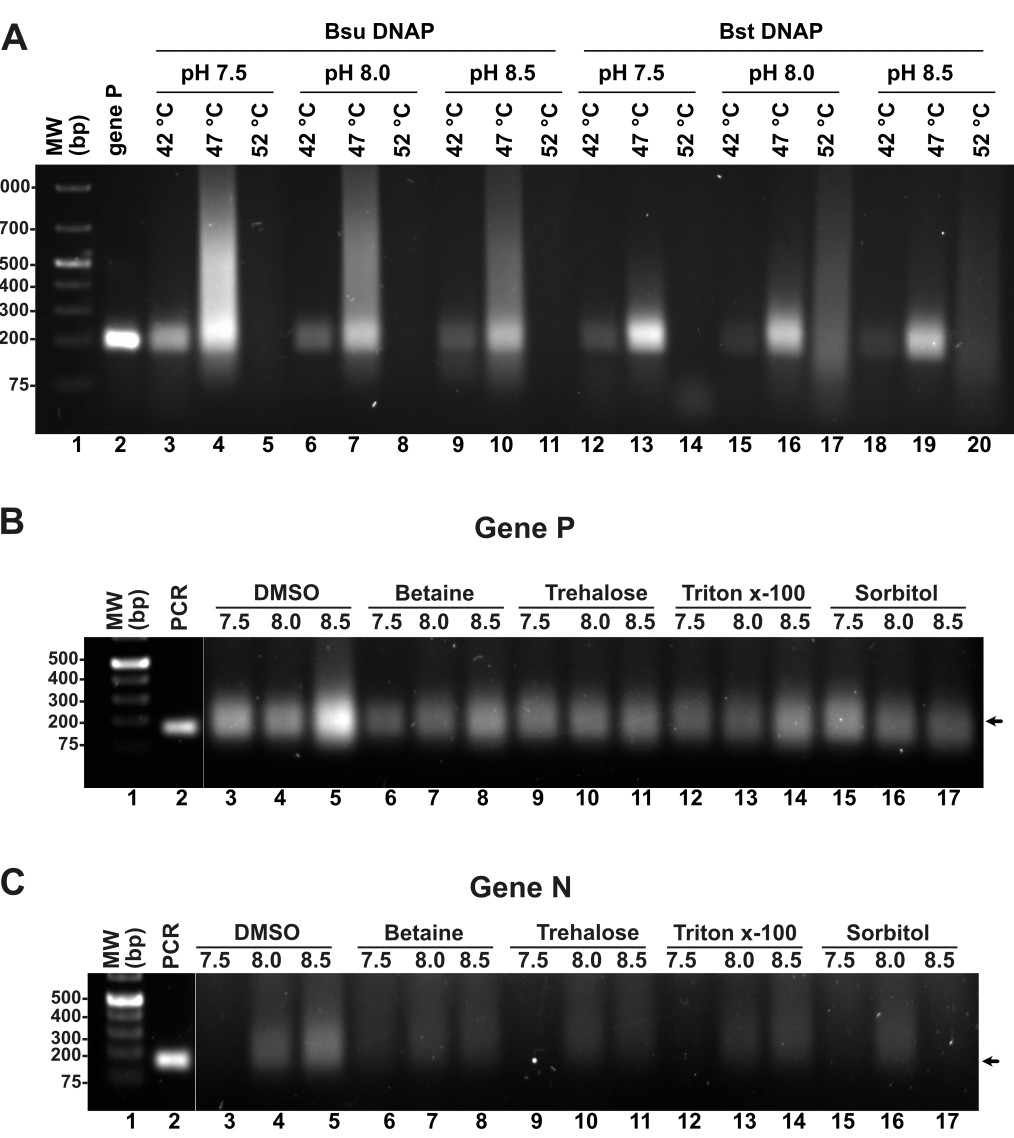

**Figure 4 Effect of temperature and additives on RPA efficiency.** (A) RPA reactions assayed at three different temperatures and pHs using *Bsu* or *Bst* DNA polymerases and visualized by agarose gel electrophoresis (B) and (C) effect of different additives on RPA reactions for genes P and N. Reactions were executed at 42 °C in the presence of *Bsu* DNAP and 80 mM of potassium acetate. Where indicated, reactions were incubated in the presence of DMSO 5%, Trehalose 2%, Sorbitol 2%, or Triton X-100 0.1%.

reactions aimed to test the detection limits using one ng of the pET28 SARS-CoV-2 plasmid of 7.3 kb that corresponds to $1.34^8$ molecules of template DNA per reaction. We use the optimized reaction and seven serial dilutions to test for the possible RPA amplification of 1.34 molecules of the genes P, E, and N, aiming to observe a signal detectable by agarose gel electrophoresis stained with ethidium bromide (Fig. 5). Our result indicate that RPA of gene P is robust as it can amplify 1.34 molecules of DNA as initial template, as a clear amplification band is observed using $1 \times 10^{-7}$ ng of plasmid DNA. When the same experiment was repeated using primers that amplified for genes E and P, the detection

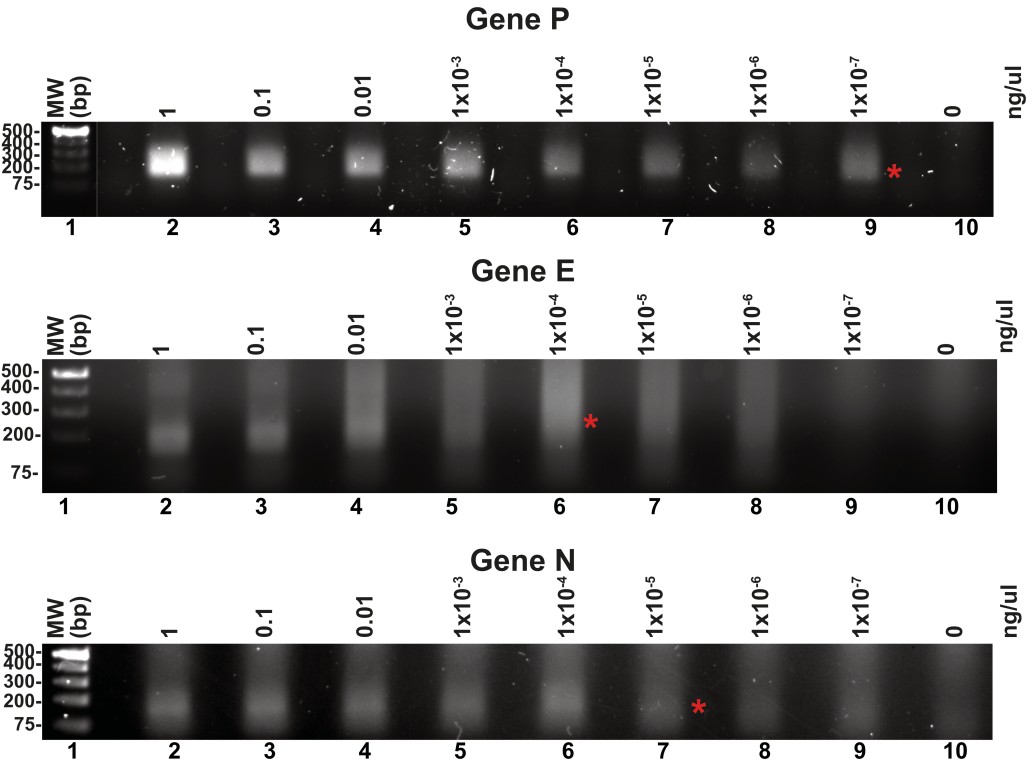

**Figure 5 Serial dilutions of RPA reactions.** RPA reactions using 10-fold serial dilution from one ng to $1 \times 10^{-7}$ ng of plasmid DNA for genes P, E, and N respectively and their corresponding negative controls. The RPA amplifications products were run on a 1.5% agarose gel electrophoresis and visualized with ethidium bromide. A red asterisk indicates the presence of a sharp well-defined band that we interpreted as the detection limit by this method.

limit corresponds to $1 \times 10^{-4}$ and $1 \times 10^{-5}$ ng per reaction respectively, as a sharp band is observed at those DNA concentrations (Fig. 5).

## Homemade RPA can detect RNA genomes by synthesizing cDNA *via* Reverse Transcriptase using patent-free enzymes

A multitude of diseases including COVID-19, dengue or chikungunya are originated by RNA viruses. Thus, many important genomes consist of RNA. This molecule is intrinsically labile and is not a substrate for canonical DNA polymerases. Thus, a home-made detection system must contain a step for the conversion of RNA to cDNA. This reverse transcription reaction is executed by several reverse transcriptases that are active at temperatures higher than 55 °C, a temperature that helps denature RNA secondary structures. In our home-made system we proposed to use a thermoresistant Moloney murine leukemia virus (MMLV) reverse transcriptase (RT) dubbed MMLV RT-4M (*Okano et al., 2017*). MMLVRT-4M is a quadruple mutant of wild-type MMLV RT devoid of RNase H activity and with thermotolerance to 60 °C (*Okano et al., 2017*; *Yasukawa et al., 2010*). To initially test the efficiency of MMLV RT-4M, we synthesized transcripts of N, E, and P gene using T7 RNA polymerase. Transcription reactions using linearized and supercoiled templates

produced RNA transcripts accordingly to the distribution and molecular mass of the three T7 RNA promoters and terminator sequences in the modified pET28b plasmid (Figs. 6A and 2A). The number of *in vitro* synthesized transcripts on a linearized template increases at least 10-times in comparison to the number of synthesized transcripts using a supercoiled template, presumably due the rapid turnover of T7 RNA polymerase on linearized templates (Fig. 6A, lanes 5 and 9). RNA transcripts were subject to DNAse I treatment and gel purified to be used as a template for cDNA synthesis. Purified RNAs were subject to Reverse transcription reactions to test the efficiency of the MMLRT-4M (Fig. 6B). During RT reactions we observed the apparition of a band on an agarose gel that corresponds to cDNA synthesis by MMLRT-4M, indicating the success of the reverse transcription of gene N (Fig. 6B, lane 5). To selectively detect an amplified region in RPA reactions, a labeled oligonucleotide conjugated with fluorescein that contains an abasic site near its central position and a 3′-end occupied by a nucleotide in the form of a dideoxide, is commonly used. Selectivity in RPA to avoid false negatives is achieved by the activity of an *E. coli* gene dubbed Endonuclease IV (nfo) (*Cunningham et al., 1986*) (reviewed in *Chen & Xia, 2022*) that cleaves the abasic site and the 3′-end dideoxide creating a new 3′-end OH for chain growth. Thus, the hybridization of the probe serves as a substrate for an abasic site DNA glycosylase that cleaves the abasic site and removes the 3′-end dideoxynucleoside and permits the growth of the strand. To investigate the potential use of a DNA glycosylase in our RPA assay we synthesized and oligonucleotide harboring an abasic site a 5′-FITC and a 3′-end chain terminator that is complementary to gene N of SARS CoV2 (Fig. 6C, lane 1). This oligonucleotide can only be extended when hybridized to a cDNA region and cleaved. We executed reactions in the presence of PCR and RPA amplified segments of gene N using *Thermus thermophilus* Endo IV, an enzyme that shares 37% amino acid identity with *E. coli* IV (nfo) (*Li et al., 2018*). In our reactions we observed that *T. thermophilus* Endo IV cleaves the 5′ FITC and 3′-end chain terminator probe and that this probe can be extended in the presence of B*su* DNA polymerase for both PCR and RPA substrates (Fig. 6C, lanes 2 to 5).

## Homemade RPA can detect the presence of SARS-CoV-2 from patients by lateral flow immunoassay (LFD) using cDNA samples amplified in a clinical setting

RPA products, antigens, and antibodies can be detected using lateral flow tests (LFD). LFDs are immunoassays that are visualized with the naked eye on a test strip. The lateral flow strip adapted to visualize amplified DNA products is based on the ability of oligonucleotides to be conjugated at their 5′ end by a marker such as fluorescein (FITC) and biotin (*Wang et al., 2021*). Thus, in principle by placing the RPA reaction on the lateral flow strip, the sample containing the amplified SARS-CoV-2 DNA would migrate by capillary action through a region containing an anti-biotin antibody that traps the amplified DNA strand (Fig. 7A). In order to investigate if our "home-made" system would be able for LFD detection, we proceed to execute an RPA amplification of eight clinical samples of SARS-CoV-2 (four positive and four negative) using mRNA that was isolated from clinical samples from a previous study and converted to cDNA in a diagnostic dedicated laboratory (*Tapia-Sidas*

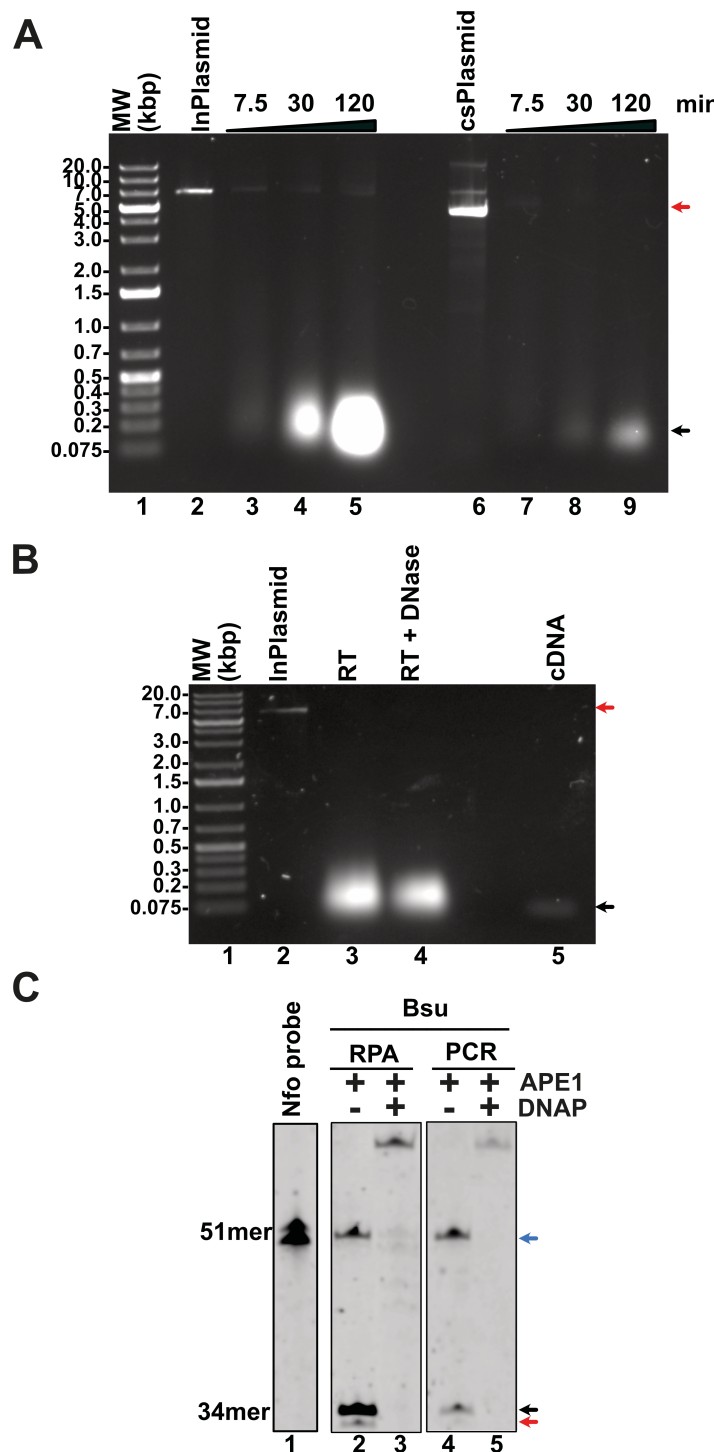

**Figure 6 Evaluation of transcription, reverse-transcription, and selectivity test for of SARS-CoV-2 detection by RPA.** (A) Transcription reactions of SARS-CoV-2 genes RpRd, E, and N by home-made T7RNAP on linear and supercoiled plasmids (lanes 3 to 5 and 7 to 9, respectively). (B) Purification of the transcript of gene N from SARS-CoV-2 transcripts. (continued on next page...)

**Figure 6 (…continued)**
Initial transcript of gene N and its DNAse I treatment (lanes 3 and 4). Amplification band showing cDNA synthesis of gene N of S SARS-CoV-2 by MMLV RT-4M (lane 5). (C) APE1 nuclease processing and extension products by Bsu DNA polymerase. Untreated probe harboring an abasic site (lane 1) treated with *T. thermophilus* Endo IV that cleaves the probe when is hybridized to a PCR or RPA amplification product of gene N (lanes 2 and 4). In the presence of Bsu DNA polymerase, the cleaved probe can be used as a primer for DNA amplification and produced amplification products that are longer than the initial probe (lanes 3 and 5).

*et al., 2023*). cDNA samples were purified in a diagnostic setting and tested as positive or negative using end point RT-PCR. In this experiment, we repeated our RPA reaction using cDNA and three oligonucleotides: a biotin 5′ end oligonucleotide and an unlabeled oligonucleotide that amplify for a region of 150 bp using the four positive and four negative samples. Reactions were incubated for 20 min and separated on an agarose gel (Fig. 7B, lanes 2 and 3 and 8 and 9). Lanes 2 and 3 of Fig. 6B correspond to two positive samples and lanes 8 and 9 to two negative samples previously tested by end-point RT-PCR. The presence of amplification bands in samples run in lanes 2 and 3 and the lack of amplification in samples that tested negative for the presence of SARS-CoV2 indicated the feasibility to use RPA to detect SARS-CoV2. To further expand this observation, we used a 1:50 dilution of each RPA positive sample and performed RPA experiments using the abasic site probe and biotin 5′ end oligonucleotide. In this experiment we added *T. thermophilus* APE1 endonuclease to the reaction for specific amplification. Reactions were incubated for another 30 min and separated on an agarose gel (Fig. 7B, lanes 4 to 7). Three of the four positive samples generated a band of lower molecular mass that corresponds to the amplification product of the abasic site processed oligonucleotide, and the forward biotinylated primer and one sample generated a barely visible amplification product (Fig. 7B, lanes 4 to 7, red asterisk). In contrast, the four negative samples did not produce an amplification product (Fig. 7B, lanes 8 and 9 and data do not show). We ran the four samples identified as positives by RT-PCR and observed detection in the LFD assay. In contrast, the negative samples did not react in the LFD assay (Fig. 7B, lanes 8 and 9).

## DISCUSSION

Open-source initiatives to develop an infrastructure for detecting human diseases surged during the COVID-19 pandemic. Multiple examples of diagnostic initiatives obtaining the necessary enzymes for RT-qPCR and LAMP by recombinant methods are available in the literature (*Cerda et al., 2024*; *Kellner et al., 2022*; *Mascuch et al., 2020*; *Matute et al., 2021*; *Tapia-Sidas et al., 2023*; *Yip et al., 2022*). The seminal recombinase polymerase amplification (RPA) manuscript by Piepenburg and coworkers rely in purified T4 recombination proteins by the authors (*Piepenburg et al., 2006*). However, in most cases, the use of RPA technology relies on commercially available kits (*i.e.,* https://www.twistdx.co.uk/rpa/; https://www.igeneye.com/; https://www.amp-futurebio.com/). Notably, pioneering efforts by the Yasukawa group have highlighted the potential of T4 homologous recombination (HR) enzymes for application in RPA (*Juma et al., 2021*; *Juma et al., 2022*; *Kojima et al., 2021*). Additionally, other critical enzymes necessary for

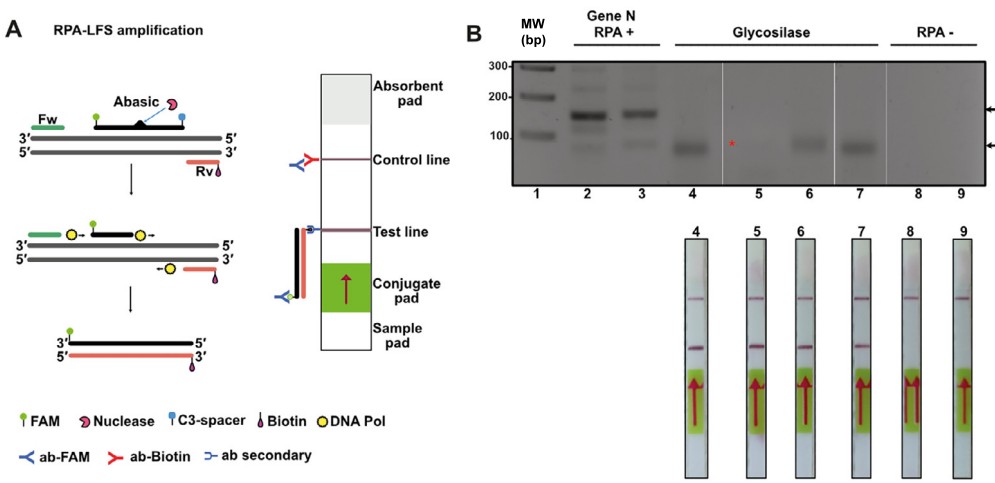

**Figure 7 Lateral flow immunoassay (LFD) using the open source T4-HR system.** (A) Graduated lateral flow strip to detect amplified SARS-CoV-2. The sample is deposited on the sample pad containing a nitrocellulose membrane. The sample migrates by capillary action, and in our case the double-stranded DNA has a biotic molecule attached at its 5′ end and the probe a fluorescein molecule also at the 5′ end. The lateral flow graduated strip contains colloidal gold-conjugated anti-fluorescein antibodies and control antibodies. The presence or absence of amplified DNA is made visually in the control band or line. (B) RPA reaction from SARS-CoV-2. The first four strips correspond to samples from positive patients (previously identified by end-point RT-PCR) and the last two are negative controls. In the positive strips there is an amplification of the SARS-Cov2 N gene that is observed in as a line in the corresponding position.

constructing the RPA system, including creatine kinase, DNA polymerases, and APE-1 endonucleases, are accessible as open-source, patent-free reagents (Table S1). In this study, we aim to integrate these efforts to develop an open-source RPA system that incorporates the T4 HR system, DNA polymerases, and specialized enzymes.

Here we found two key variables to increase the amount of amplified RPA product and eliminate primer-dimers: (1) executing RPA at temperatures greater than 37 °C and (2) increasing the molar concentration of gp32. The T4 HR system is forced to be assayed at temperatures that do not exceed 45 °C because of the limited stability of mesophilic T4 HR enzymes. In our hands, the addition of DMSO and 80 mM of potassium acetate are also key determinants to increase RPA specificity and efficiency. High concentration of the osmolyte betaine (from 0.3 to 0.9 M) result in greater amounts of RPA products and an increase in specificity (*Luo et al., 2019*) and this osmolyte can be used in future rounds of optimization of this system.

## CONCLUSIONS

In conclusion, our system provides a versatile platform for exploring a wide range of conditions in RPA assays. Despite RPA's effectiveness as a tool for nucleic acid–based molecular diagnostics, the dependency on commercial kits restricts the ability to precisely control enzymatic conditions. By making enzymatic components readily accessible, our approach enables systematic investigations to mitigate nonspecific amplification and improve detection sensitivity in RPA. Moreover, these methods can be efficiently deployed

in regions with moderate infrastructure for recombinant protein production and seamlessly adapted for tracking emerging diseases, all within an open, patent-free framework. The plasmids that code for protein products used in this study were deposited in Addgene (pET19b-pss-gp32, Plasmid # 236044; pET19b-pps-UvsY, Plasmid # 236045; pET19b-pps-UvsX, Plasmid # 236046; pCOLDI-KF-*Bst* DNApolI, Plasmid # 236047; pET19pps-EndoIV (*Thermus thermophilus*) Plasmid # 236048).

## ACKNOWLEDGEMENTS

We thank Drs. Kiyoshi Yasukawa for plasmids overexpressing MM4-MMLV-RT.

### Funding

This work was supported by the CONACHYT-COVID emergence grant # 311960, CONACHYT-NFRAESTRUCTURA grant #317147, and grant AMEXCID-2020 "Fondo Binacional México-Uruguay". FCA, APC, NBT, & CR were also the recipients of CONACHYT-Mexico graduate study fellowships. The funders had no role in study design, data collection and analysis, decision to publish, or preparation of the manuscript.

### Grant Disclosures

The following grant information was disclosed by the authors:
The CONACHYT-COVID emergence grant: # 311960.
CONACHYT- NFRAESTRUCTURA: #317147, AMEXCID-2020.
FCA, APC, NBT, & CR were also the recipients of CONACHYT-Mexico graduate study fellowships.

### Competing Interests

Rogerio R. Sotelo-Mundo is an Academic Editor for PeerJ.

### Author Contributions

- Francisco Cordoba-Andrade conceived and designed the experiments, performed the experiments, analyzed the data, prepared figures and/or tables, authored or reviewed drafts of the article, and approved the final draft.
- Antolin Peralta-Castro performed the experiments, analyzed the data, prepared figures and/or tables, authored or reviewed drafts of the article, and approved the final draft.
- Paola L. García-Medel performed the experiments, prepared figures and/or tables, and approved the final draft.
- Eduardo Castro-Torres performed the experiments, prepared figures and/or tables, and approved the final draft.
- Rogelio Gonzalez-Gonzalez performed the experiments, prepared figures and/or tables, and approved the final draft.
- Atzimba Y. Castro-Lara performed the experiments, prepared figures and/or tables, and approved the final draft.

- Josue D. Mora Garduño performed the experiments, prepared figures and/or tables, and approved the final draft.
- Claudia D. Raygoza performed the experiments, prepared figures and/or tables, and approved the final draft.
- Noe Baruch-Torres performed the experiments, analyzed the data, prepared figures and/or tables, and approved the final draft.
- Alejandro Peñafiel-Ayala performed the experiments, analyzed the data, prepared figures and/or tables, and approved the final draft.
- Corina Diaz-Quezada performed the experiments, prepared figures and/or tables, and approved the final draft.
- Cesar S. Cardona-Felix conceived and designed the experiments, analyzed the data, prepared figures and/or tables, and approved the final draft.
- Fernando Guzman Chavez conceived and designed the experiments, analyzed the data, prepared figures and/or tables, and approved the final draft.
- Carlos H. Trasviña-Arenas conceived and designed the experiments, analyzed the data, prepared figures and/or tables, and approved the final draft.
- Rogerio R. Sotelo-Mundo conceived and designed the experiments, analyzed the data, prepared figures and/or tables, and approved the final draft.
- Beatriz Xoconostle-Cazares conceived and designed the experiments, analyzed the data, prepared figures and/or tables, and approved the final draft.
- Agustino Martínez-Antonio conceived and designed the experiments, analyzed the data, prepared figures and/or tables, and approved the final draft.
- Luis Gabriel Brieba de Castro conceived and designed the experiments, performed the experiments, analyzed the data, prepared figures and/or tables, authored or reviewed drafts of the article, and approved the final draft.

## Human Ethics

The following information was supplied relating to ethical approvals (*i.e.*, approving body and any reference numbers):

ETHICS COMITE FOR RESEARCH IN HUMAN BEINGS (COBISH)-CINVESTAV, Protocol number: 062/2020.

## Data Availability

The raw data is available in the Supplementary Files.

## Supplemental Information

Supplemental information for this article can be found online at http://dx.doi.org/10.7717/peerj.19758#supplemental-information.

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
