# Peer review of "The concentration of single-stranded DNA-binding proteins is a critical factor in recombinase polymerase amplification (RPA), as revealed by insights from an open-source system"

_PeerJ, doi:10.7717/peerj.19758_

## Round 0.1 · original submission · Major Revisions

Please address all the reviewer comments with special emphasis on the first reviewer's comment on essential detail missing in the protocol.

Reviewer 1 ·

Basic reporting

The paper describes the expresion and full assembly of all 7 proteins needed for
an RT-RPA system. In contrast to others the authors also expressed Bst DNA polymerase,
creatine kinase, thermostable MMLV reverse transcriptase.

They succesfuly demonstrate showing the activity of the enzymes and demonstarte
this by testing SARS-CoV-2 assays with the new RPA mix.

The paper is will written and designed and all data presented are supported by the evidence provided.
The paper however fails to explain essential details of the protocol (see below).

Experimental design

Introduction and line 217:

The role of UvsY is not properly explained.
Given the fact that you find the role of UvsY important, which I can confirm
from our own experimetation, I think there should be a more detailed explanation
of the UvsY function.

Mat & Meth

line 128: Please explain what you mean by the Berlin protocol as your reference
further down in the text (which should be moved to here) merely links to the SARS-CoV-2
reference sequence.

Validity of the findings

Results and Discusiion

One of the key findings of the Kojima publcation protocol was that proteins were stored in
a traditional Borate buffer witout glycerol and assembeled into an RPA mix. Our
own excperiments show that assembling 7 proteins containing glycerol does not
generate an active RPA mix. In recognition of this fact even company
IntactGenomics which was the first to over all RPA proteins on the free
market now offers them without glycerol. Please explain how you dealt with the
glyceol issue and discuss.

For the ambition to publish an open source protocol I would suggest to include a table that
shows the final composition of the proteins and buffer mixtures for both types of assays prsented.

Reviewer 2 ·

Basic reporting

Addressed below.

Experimental design

Addressed below.

Validity of the findings

Addressed below.

Additional comments

This manuscript from Cordoba-Andrade and coworkers recreates the well-known and widely used recombinase polymerase amplification (RPA) reaction using an "open-source" system with homemade components. This is an important paper and I believe it will be widely used by the community. This is because RPA is widely used in a range of diagnostic technologies. But at present, the RPA components are only available from a sole-source supplier in the UK that has experienced intermittent supply problems since the COVID pandemic. The paper demonstrates that the homemade components can reconstitute RPA, and does a limited-scope investigation of concentration dependence. A limited scope investigation was suitable because most of the work relating to this was done in the original Piepenburg 2016 work describing RPA for the first time. It was necessary to show that the authors' products suffice for RPA, and they have done so. As such, I recommend publication of a revision. My suggestions are in two broad categories - the first relate to some experimental concerns, and the second relate to ensuring that the necessary steps (plasmid deposition, inclusion of some experimental details) are done to ensure broad use of the important work shown in this paper.

Major:

A plasmid deposition is not given for the enzymes used. If this is an open-source system as claimed, all the plasmids should be deposited in Addgene.

Similarly, the general description of purification of reaction components given in section 2.2 is insufficient. The utility of this paper rests on it being a tool other investigators can use. It appears that most of this information is present in the supplementary information, but it was hard to find. This apparently is referred to as "Annex" several times in the text, which was a bit confusing, because it is only labeled this way in the callout in the text. In this information, it would be helpful to supply details of the transformation procedure and the dialysis (e.g., was tubing or a cartridge used, what kind, etc?).

The method of quantitation for the proteins used is not given. Given that much of the results rely on varying the concentration of RPA reaction components, this should be specified in as much detail as possible.

The original Piepenburg et al (2006) RPA manuscript is not cited until the results section. It should be cited in the introduction when RPA is introduced. Similarly, any contrasting results with this paper should be highlighted. I'm not overly concerned if they're a bit different in, e.g., concentration dependence, given the difficulty in quantitating overexpressed proteins precisely, but readers should be aware and it would be of value to show any differences.

On lines 233-237, the inclusion of a His-tag on UvsX is described as being detrimental. Including these data would be helpful.

Minor:

I was a bit disappointed that there wasn't an examination of polymer crowding agents, since the original Piepenburg manuscript showed that this was important, and this new reconstructed system could behave a bit differently. I'm not insistent it be included in a revision if the data are unavailable, but if the authors do have these data, including it would enhance the manuscript significantly.

Throughout figures, please label MW marker with units and specify meaning of lanes where unclear (e.g., "26619" in Fig 1 mw marker)

SI, "MMLV-RT 4M (mutante termoestable de MMLV)" please change the parenthetical text to English

SI, "Creatine Kinasae" is misspelled

SI section 3, typo, "RNAPase"

---

## Round 0.2 · Minor Revisions

Address the reviewer comments.

Reviewer 1 ·

Basic reporting

The authors have responded to all review comments satisfactorily, some minor editing and a new comment in the discussion remain to be adressed.

Experimental design

see abone

Validity of the findings

see above

Additional comments

none

Annotated reviews are not available for download in order to protect the identity of reviewers who chose to remain anonymous.

Reviewer 2 ·

Basic reporting

See below

Experimental design

See below

Validity of the findings

See below

Additional comments

In this revision, the authors have addressed almost all my comments. I would ask that they please add the addgene accession numbers for their plasmids to help others find them and maximize the utility of the paper (it is getting harder and harder to find plasmids by name on AddGene). Other than that last request, I recommend publication. Congratulations on a nice manuscript!

---

## Round 0.3 · Minor Revisions

Provided Addgene Plasmid IDs are not being returned in Addgene database. If you have submitted the sequence please make it publicly available.

---

## Round 0.4 · accepted · Accept

I hope that Addgene processes your sample soon and plasmids become accessible.